# Pathways to Increasing Adolescent Physical Activity and Wellbeing: A Mediation Analysis of Intervention Components Designed Using a Participatory Approach

**DOI:** 10.3390/ijerph17020390

**Published:** 2020-01-07

**Authors:** Kirsten Corder, André O. Werneck, Stephanie T. Jong, Erin Hoare, Helen Elizabeth Brown, Campbell Foubister, Paul O. Wilkinson, Esther MF van Sluijs

**Affiliations:** 1UKCRC Centre for Diet and Activity Research (CEDAR) and MRC Epidemiology Unit, University of Cambridge, Cambridge CB2 1TN, UK; andreowerneck@gmail.com (A.O.W.); styj2@medschl.cam.ac.uk (S.T.J.); erin.hoare@mrc-epid.cam.ac.uk (E.H.); helen.brown@bi.team (H.E.B.); campbell.foubister@mrc-epid.cam.ac.uk (C.F.); ev234@medschl.cam.ac.uk (E.M.v.S.); 2Department of Physical Education, Universidade Estadual Paulista “Júlio de Mesquita Filho” (UNESP), Presidente Prudente 19000-000, Brazil; 3Food & Mood Centre, Centre for Innovation in Mental and Physical Health and Clinical Treatment, School of Medicine, Faculty of Health, Deakin University, Melbourne 3004, Australia; 4IMPACT Strategic Research Centre, School of Medicine, Deakin University, Barwon Health, Geelong 3220, Australia; 5Department of Psychiatry, University of Cambridge and Cambridgeshire and Peterborough NHS Foundation Trust, Cambridge CB2 0SZ, UK; pow12@cam.ac.uk

**Keywords:** intervention, physical activity, mental health, adolescent, school, health promotion

## Abstract

We assessed which intervention components were associated with change in moderate-to-vigorous physical activity (MVPA) and wellbeing through proposed psychosocial mediators. Eight schools (*n* = 1319; 13–14 years) ran GoActive, where older mentors and in-class-peer-leaders encouraged classes to conduct two new activities/week; students gained points and rewards for activity. We assessed exposures: participant-perceived engagement with components (post-intervention): older mentorship, peer leadership, class sessions, competition, rewards, points entered online; potential mediators (change from baseline): social support, self-efficacy, group cohesion, friendship quality, self-esteem; and outcomes (change from baseline): accelerometer-assessed MVPA (min/day), wellbeing (Warwick-Edinburgh). Mediation was assessed using linear regression models stratified by gender (adjusted for age, ethnicity, language, school, BMI z-score, baseline values), assessing associations between (1) exposures and mediators, (2) exposures and outcomes (without mediators) and (3) exposure and mediator with outcome using bootstrap resampling. No evidence was found to support the use of these components to increase physical activity. Among boys, higher perceived teacher and mentor support were associated with improved wellbeing via various mediators. Among girls, higher perceived mentor support and perception of competition and rewards were positively associated with wellbeing via self-efficacy, self-esteem and social support. If implemented well, mentorship could increase wellbeing among adolescents. Teacher support and class-based activity sessions may be important for boys’ wellbeing, whereas rewards and competition warrant consideration among girls.

## 1. Introduction

Over the last century, physical activity has declined concurrently, with an increase in the burden of common mental health disorders [1]. Globally, physical inactivity is thought to cause 9% of premature deaths and is estimated to cost 53.8 billion in health care [2,3]. Due to the importance of inactivity as a health risk and the high prevalence of inactivity worldwide, one of the World Health Organisation’s nine global targets is a 10% relative reduction in the prevalence of inactivity by 2025 [4]. However, recent global data suggest that meeting this global target looks increasingly unlikely [5]. The World Health Organisation recommend that all children between 5 and 17 years old do at least 60 minutes of physical activity every day [6], which aligns with British recommendations [7]. Recent evidence suggests that worldwide, the majority of young people (81%) aged 11 to 17 years do not meet these recommendations [5]. Preventing the decline of physical activity during adolescence is a major public health priority [8], as inactivity during adolescence is likely to last into adulthood, resulting in increasing health risks [9,10].

Concurrently with physical activity declines during adolescence, this transitional life stage/phase is a period of increased risk of mental health problems [11]. Of all the mental disorders, half will have emerged by the age of 14 years [12]; in 2017, it was estimated that 24% of British girls and 9% of British adolescent boys experienced depressive symptoms [13] and suicide is one of the major causes of mortality in adolescence and young adulthood [14]. Risk factors for poor mental health consist of a broad range of individual, family, environmental, social and other factors [12]; this range of correlates is likely to contribute to the large variation in the effectiveness of trials aiming to improve adolescent mental health [15]. More effective strategies are needed to support and enable positive mental health, including wellbeing, among young people [16].

Social aspects of physical activity participation are proposed to strengthen relationship-building and other interpersonal qualities that may additionally protect against the development of mental health problems in addition to increasing physical activity levels [17]. Therefore, physical activity interventions may have a positive impact on activity levels and/or wellbeing but relatively little is known about the mechanisms by which any effects may occur [18]. Proposed mechanisms between physical activity and mental health outcomes among adolescents commonly refer to neurobiological (e.g., neurotrophic gene and protein expression, grey matter volume and activation), psychosocial (e.g., social connectedness, physical self-perceptions) and behavioural (e.g., sleep volume and quality, self-regulation) domains [19].

We used data from the 12-week Get Others Active, ‘GoActive’ intervention that was based on self-determination theory and co-designed with students and teachers using a participatory approach [20]. Co-design of interventions is recommended to improve intervention outcomes among young people [20]. The intervention components are summarised alongside supporting research evidence and the participatory rationale from the co-design process (Table 1). Briefly, the intervention involved training older adolescents to encourage Year 9 classes to choose two new weekly activities from a list of 19 available (included as Appendix A) [21] with teachers encouraged to use one tutor time per week for participation in these activities. Students gained points and rewards for activity in and out of school. The intervention used was co-designed with students and teachers; co-design of interventions is recommended to improve intervention outcomes among young people [20]. We hypothesised that intervention components suggested by students and teachers (mentorship, leadership, teacher support, class-based activity sessions, competition, rewards and online activity tracking) would influence physical activity and wellbeing through social support, self-efficacy, group cohesion, friendship quality and self-esteem. The primary outcome was average daily minutes of accelerometer-assessed moderate-to-vigorous physical activity (MVPA assessed using Axivity) at 10 months post-intervention. Although the intervention did not show a direct effect on overall MVPA (primary outcome) or wellbeing (secondary outcome) [22], particular engagement with components may be associated with mediators and outcomes, for example, encouragement provided by older adolescent mentors could be associated with increased self-efficacy and social support, which may be associated with changes in physical activity and wellbeing as proposed previously [23,24].

Our initial process evaluation results suggested that perception of, and engagement with, different components may have had a potentially differential impact on the proposed mediators [25]; whether or not this is also associated with an effect on physical activity or wellbeing is useful to inform future intervention design. The components used in GoActive (particularly mentorship and class-based activity sessions) are commonly used in physical activity promotion but little is known about the mechanisms by which they may increase outcomes such as physical activity and mental wellbeing.

We aimed to apply mediation analysis in a novel approach to evaluate the potential mediating role of psycho-social factors (social support, self-efficacy, group cohesion, friendship quality and self-esteem) in the association between engagement in intervention components suggested by students in our intervention co-design process (mentorship, leadership, class-based activity sessions, competition and rewards) with changes in physical activity and wellbeing.

## 2. Methods

### 2.1. Study Design

The two-arm parallel-group cluster randomised controlled trial (RCT) consisted of a 12-week intervention phase with main outcome assessment at 10-month follow-up [21]. Ethical approval was obtained from the University of Cambridge Psychology Ethics Committee (PRE.126.2016). Passive opt-out parental consent was sought and written student assent was obtained. The study was prospectively registered as ISRCTN31583496.

### 2.2. Participant Recruitment and Inclusion Criteria

All state-run secondary schools in Cambridgeshire and Essex (*n* = 103) were invited into the study between April and July 2016. The first 16 schools to agree to participate were included and provided school level written informed consent; all students in Year 9 in participating schools during the 2016–2017 academic year were eligible for inclusion. The school year in British non-fee-paying schools usually runs from early September to the third week of July. There are holidays at Christmas and Easter (approximately two weeks each) with ‘half-term’ (one-week holidays) in late October, mid/late February and late May. Baseline assessment took place early in Year 9 (September 2016–January 2017) when participants were aged 13–14 years. After baseline, participating schools were randomised to intervention or no-treatment control arms; allocation used a randomisation list prepared in advance by the trial statistician independent from the measurement team using a random number generator in Stata. Randomisation was stratified by school-level pupil premium (below or above the county-specific median) and county (Cambridgeshire or Essex). Pupil premium, used as a proxy for school level deprivation, is school funding aiming to reduce effects of deprivation [26]. Only data from schools randomised to the intervention (*n* = 8; *n* = 671 participants) were included in this analysis.

### 2.3. Intervention

The GoActive intervention was developed following an evidence-based iterative approach in which we incorporated existing evidence and qualitative work with adolescents and teachers [20]. The student and teacher rationale for the components included is summarised in Table 1. GoActive aimed to increase physical activity through increased social support, self-efficacy, group cohesion, friendship quality and self-esteem and was implemented in tutor groups using a student-led tiered-leadership system. The 12-week intervention trained older adolescent mentors and in-class-peer-leaders to encourage classes to select two new weekly activities with one tutor time a week to participate in these. Students gained points and rewards for activity in and out of school. During the first 6 weeks of the intervention, a facilitator (health trainers employed and funded by local councils) was provided; during the second 6 weeks, external support for the programme was reduced to encourage school-led sustainability. Intervention facilitators provided school teachers and older mentors with training, support and resources for intervention delivery. Mentorship and peer-leadership had high acceptability in co-design work with students and were also suggested by teachers to address time pressures stated by teachers in our development work as a barrier to participation in activity promotion programmes [20]. In addition, between-class competition was incorporated as a strategy to encourage teacher enthusiasm, with teachers suggesting that they were often very competitive with other tutor teachers within the same year group [20].

### 2.4. Measures

Outcome assessments using largely identical procedures were undertaken at baseline and post-intervention (14–16 weeks post-baseline) in the school. Quantitative process evaluation data were collected in post-intervention questionnaires adapted from those used in the feasibility study (available as Appendix A) [20]. Trained research staff conducted measurements using standardised protocols and instruments as detailed in the protocol [21].

### 2.5. Outcomes

#### 2.5.1. Physical Activity (Accelerometry)

The primary outcome was change in overall minutes of MVPA (post intervention minus baseline), measured using wrist worn activity monitors (Axivity) assessing acceleration (continuous waveform data). Participants were asked to wear the monitors for 7 days continuously, worn for 24 hours a day on their non-dominant wrist. These monitors have been validated to assess physical activity energy expenditure [41]. Monitor output was processed to provide minutes spent in MVPA to be equivalent to ≥2000 ActiGraph cpm [21]; further details on accelerometer data processing can be found elsewhere [22].

#### 2.5.2. Wellbeing

The Warwick Edinburgh Mental Wellbeing Scale was used to assess mental wellbeing [42]. The 14-item scale asks participants to indicate what best describes their experiences of a collection of statements over the past two weeks. Items relate to both hedonic and eudaimonic experiences of mental health including positive affect (e.g., ‘*I’ve been feeling optimistic about the future’*), relationships (e.g., ‘*I’ve been feeling close to other people’*) and emotional functioning (e.g., ‘*I’ve been dealing with problems well’*), each rated on a 5-point Likert scale is used with responses ‘*none of the time’*, ‘*rarely*’, ‘*some of the time’*, ‘*often*’, and ‘*all of the time’* scored 1 to 5 respectively. The scale has shown good content validity and correlates well with other mental health and wellbeing scales, including the Positive and Negative Affect Scale [43,44], Short Depression-Happiness Scale [45], the World Health Organisation—Five Well-Being Index [46]. Cronbach’s alpha has been shown to be 0.89 among a student sample and 0.91 in a population sample [42]. Overall, the scale has shown good internal consistency, good test-retest reliability and good face validity [47]. Participants who responded to all 14 items on the scale were included in this study, and their score was the average of the item responses (out of possible 1 to 5).

### 2.6. Exposures: Engagement with Intervention Components

Data on engagement with intervention components were collected using post-intervention questionnaires and questions were adapted from those used in the feasibility study [20]. To assess engagement with mentors, students were asked, “Within the GoActive programme, Mentors (a) motivated me to be active, (b) were enthusiastic about GoActive, (c) offered lots of different activities to take part in, (d) came in to run GoActive almost every week and (e) explained activities clearly. Students were also asked to assess engagement with their form tutors (teachers), “Within the GoActive programme, Teachers (a) motivated me to be active and (b) were enthusiastic about GoActive. Response options were from 1 “Strongly agree” to 4 “Strongly disagree”. Items were reverse-coded so higher values were indicative of more positive responses and a mean calculated. Perception of other intervention components was assessed using Likert scales (‘Do not like it at all’ (1) to ‘Like it a lot’ (5)); these were class-sessions (mean of trying new activities and using tutor time), rewards (individual prizes), competition (mean of class and individual competition) and peer leaders (mean of reported leadership and engagement). Online activity tracking was assessed using the points entered on the study website; participant scores were dichotomised as having entered points versus not entering points on the website.

### 2.7. Potential Mediators

Questionnaires were used to assess potential mediators at baseline and post intervention. Social support for physical activity was a mean derived from 9 self-reported items (response range 1–4) from the European Youth Heart Study [48]. Self-efficacy was a mean of eight self-reported items from Reynolds’ Psychosocial Predictors of Physical Activity: Self-efficacy scale [49] (response categories 1–6). Group cohesion was assessed by an adapted social network modelling tool in which participants were provided with a list of tutor group members on a laptop and were asked to select up to five names of their friends from the list provided. These data were used to derive in-degree (the number of people identifying the participant as a friend) and out-degree (the number of friends that participant lists as a friend) [50]. Friendship quality was a mean score of eight self-reported items used in the ROOTS project (equally weighted) [51], with a response range of 1–5. Self-esteem was a mean score of self-reported items using the 10-item Rosenberg Self-Esteem Scale [52], which had original response options 1–4. For each potential mediator, change was calculated as post-intervention minus baseline.

### 2.8. Potential Confounders

Data on student age, sex, ethnicity, BMI and family socioeconomic position (SEP) were derived from self-report questionnaires at baseline. Ethnicity was self-reported by participants who were given 20 response options and additional free text completion options, which were collapsed into five categories for descriptive purposes and dichotomised for pre-specified moderation analyses (‘White’ versus remaining categories). Participants completed six items from the Family Affluence Scale relating to family car ownership, holidays, computers, availability of bathrooms, dishwasher ownership and having their own bedroom, which was used as a proxy of individual socio-economic position by summing answers (possible range 0–13), and dividing into predefined groups (i.e., affluence: low = 0–6, medium = 7–9, high = 10–13) [53].

Anthropometry (height, weight, waist circumference, bio-impedance) was assessed at baseline and 10-month follow-up by trained staff; Body Mass Index (BMI) z-score was calculated from height, weight, age and sex [54].

### 2.9. Statistical Analysis

Characteristics of the sample were described using values of mean, standard deviation and frequency. Sex differences in participant characteristics, change in outcomes, change in potential mediators and satisfaction of intervention components were tested using the Mann–Whitney U test.

Data are included from measurements at baseline and post-intervention. Due to initial process evaluation findings indicating differences in intervention acceptability for boys and girls, all analyses were stratified by gender [25]. Linear regression models, adjusted for age, ethnicity, language spoken at home, school (categorical variable), BMI and baseline values (for change variables) were used to examine associations between exposures, mediators and outcomes.

Mediation was assessed using three stage linear regression models according to previously described concepts [55]. Firstly, associations between exposures (intervention components) and potential mediators were assessed individually. Secondly, the association between exposures and outcomes (without mediators) was assessed to generate the ‘total effect’. Finally, the association of both exposure and mediator with outcome was assessed, generating the controlled direct effect and consequently, the natural indirect effect.

With the total and controlled direct effect, the natural indirect effect (mediation effect) was estimated and tested by generating 95% confidence intervals estimated though the bootstrap re-sampling method, which estimates random sampling with replacement. In the bootstrap resampling method, each observation has an equal chance of being included each time, so there are observations that can be included more than once and some observations are not included. We conducted 1000 bootstrap re-samplings for each estimation. Confounders were included in the models as covariates.

The bias of potential unobserved/unmeasured confounders was estimated by using an “E-value”, [56], which is defined as the minimum strength of association (risk ratio scale) that an unmeasured confounder would need to have with both exposure and the outcome to fully explain the specific exposure-outcome association, conditional on the measured covariates.

All analyses were performed using Stata version 15.1 [57].

## 3. Results

Of 3405 Year 9 students eligible for inclusion across all participating schools, 2862 (84.1%) consented, of which 1543 participants attended the eight intervention schools. Of these, 1166 (75.6%) attended a post-intervention measurement session and of those, 671 (57.5%) provided data on both outcomes (accelerometer assessed physical activity and wellbeing) and all potential mediators at both baseline and post intervention (Table 2).

Table 3 shows the associations between each exposure (intervention component) and potential mediator individually with both outcomes. Associations between intervention components and potential mediators are shown in Table 4. Although perceived teacher support and perception of rewards were directly positively associated with MVPA among boys, no potential mediators were associated with MVPA. Among girls, no exposures or potential mediators were associated with MVPA. Various intervention components and proposed mediators were associated with increased wellbeing. The variables identified differed for boys and girls.

The results of the mediation models are displayed in Figure 1, Figure 2, Figure 3 and Figure 4. Among boys, higher perceived teacher support was associated with increased wellbeing via increased social support (Figure 1). In addition, higher perceived mentor support was associated with increased wellbeing via increases in social support, self-esteem and self-efficacy (Figure 1). For boys, a higher perception of class-based activity sessions was associated with increased wellbeing via self-esteem, social support and friendship quality (Figure 2). Among girls, higher perception of mentor support was positively associated with increased wellbeing via increased self-esteem and increased social support (Figure 3). Perception of both competition and rewards was associated with increased wellbeing via self-efficacy, self-esteem and social support, but only among girls (Figure 4).

## 4. Discussion

Our results suggest that mentorship by older students within a school has the potential to increase wellbeing through increased social support and self-esteem among adolescent boys and girls. The results of the remaining mediation models differ for boys and girls. Teacher support and class-based sessions may be important if aiming to increase wellbeing among boys, whereas competition and rewards are worth further investigation among girls. No evidence was seen to support the use of online activity tracking or peer-leadership. Common intervention components used in physical activity interventions and suggested by students and teachers in co-design work, such as mentorship from older adolescents, may be suitable for use in interventions aiming to target wellbeing. No mediation models were identified for MVPA, which was the primary outcome of the intervention.

The association between perceived mentorship support from older adolescents and wellbeing aligns with our initial process evaluation results addressing adolescent perspectives of the intervention, where mentorship had the ability to foster both peer-supportive environments and those which were not conducive to activity [25]. It is likely that these activity-conducive, or thwarting, environments could influence wellbeing through the same mediators proposed for physical activity; our results suggest that the associations appear stronger with wellbeing than physical activity. This also supports other work suggesting that there may be potential for change in mental health to impact physical activity as a disengagement in everyday activities and the loss of enjoyment that characterises poor mental health is been proposed to reduce future physical activity [58,59,60]. It is less commonly explored how mental wellbeing (as opposed to depression) may influence physical activity; the direction of causality between wellbeing and physical activity remains as changes in physical activity have been proposed to lead to improvements in mental wellbeing [58,61,62]. It would be valuable to further examine the direction and strength of these associations, particularly among adolescents and using accelerometer-assessed physical activity.

It should be noted that all of the effect sizes presented here are relatively small and as it was not possible to derive standardised coefficients for the mediation models, it makes it challenging to easily compare the strength of the different associations. The coefficients for change in wellbeing roughly align with group level change in wellbeing with an average score of −0.03 (0.79) and −0.11 (0.72) for boys and girls, respectively. This average score can be translated to decreases of 0.5 and 1.5 on the total Warwick–Edinburgh score scale, which ranges from 14 to 70, with the cut-off for probable depression as 40 or below [63]. Despite small effect sizes, e-values, which represent the minimal strength necessary to fully explain the association, revealed that a relative risk between 1.18 and 1.51 would be necessary with factors inside the model for a potential confounder to remove the indirect effect. Despite these modest coefficients, these findings support further exploration of using these intervention components when targeting wellbeing in school-based interventions, as this change appears relevant on a population level. However, these findings should be taken in the context of the null results for the primary outcome of MVPA, and wellbeing being a secondary outcome in this trial.

Mentorship was strongly suggested in the co-design phase and was associated with increased wellbeing, but not physical activity, among both girls and boys. Both increased self-esteem and social support appeared to mediate the association between higher perceived mentor support and increased wellbeing. It has been previously suggested that mentors who represent, advance, create and embed a shared sense of social identity can aid participation in physical activity [64]. Even though this was not supported here regarding MVPA, our results suggest that this may extend to wellbeing. The ‘quality’ of mentorship may be particularly important when considering the potential impact on psychosocial factors, as mentorship may only work successfully if the recipients are satisfied with the mentorship on offer [25]. Therefore, the importance of recruiting appropriate mentors and of ensuring high quality consistent ongoing training is crucial [65]. The logistical challenges of mentor recruitment and scheduling training became apparent after the co-design phase as this was not perceived to be a challenge by students or teachers during participatory design work. Due to the primary aim of the programme being to increase physical activity, mentors were often chosen by schools as students who were ‘sporty’, whereas focusing on a mentor’s inter-personal skills, social standing and approachability may have been more important. Future studies may consider a detailed co-design phase focusing on practicalities of implementation, including organisation of training and facilitator retention. More work is needed to refine ongoing training and implementation procedures for using mentorship in behaviour change interventions. It is also important to conduct honest and rigorous process evaluations (including observations) in order to better establish the underlying barriers and facilitators to using mentorship in a school setting, which would also provide pointers for stages to include participatory design work in the future.

Aside from mentor support, the results differed for boys and girls, with teacher support and class-based activity sessions identified as important for boys, whereas rewards and competition were identified for girls. These gender differences could, at least partly, be explained by differences in attitudes to physical activity among boys and girls, the complex nature of girls’ relationship with physical activity and the gendered societal pressures and expectations that can enable or inhibit physical activity [66,67]. We conducted our co-design work with 26 students, 18 boys and eight girls; it is possible that they had particular views which did not represent the majority of those in the full trial [20]. Gender differences in our results may perhaps have been exacerbated by the larger amount of boys included in the co-design phase [20]. Our mediation results suggest that teacher encouragement of physical activity and co-educational class-based sessions could be more appropriate for boys. It is possible that girls may not have felt comfortable co-participating in activities, as PE in the UK is usually segregated by sex and girls may be particularly self-conscious of doing physical activity [68]. Body image is a strong predictor of MVPA in both boys and girls [69] and is also linked to wellbeing [70]. Concerns around masculine and feminine ideologies of health-related behaviour and body image are very relevant to physical activity promotion in a school setting but are rarely central to physical activity provision in schools. It follows that considering social identity in physical activity promotion may have particular value in schools as it has been proposed to have the potential to facilitate the promotion of exercise behaviour, to impact physical activity norms positively and this also has been identified as facilitating successful mentorship [64]. We previously proposed that boys’ opinions may hold more weight in the class environment [25]. If this is the case, class-based activity sessions may not have fostered increased social support among girls, and could be a potential explanation for why the class-based activity sessions and perceived teacher support were only identified as mediators among boys. This programme was designed as a whole school approach to avoid separating activity programmes by potentially sensitive characteristics (such as gender or weight status), which may lead to stigmatisation of already marginalised groups [71] and should be addressed carefully. There is a risk of physical activity promotion programmes perpetuating inequalities, such as those regarding gender, race or socio-economic status [72]; intervention design and implementation should take this into account and examine whether interventions exacerbate any existing inequalities.

Competition and rewards were only identified to increase wellbeing via self-efficacy, self-esteem and social support among girls. This is contrary to what was expected based on the literature in that girls are often stated not to like competition [73]. However, this aligns with our co-design work refining our intervention, where we conducted individual interviews with five students (three girls) who did support the competition element [20]. One additional possible explanation for the differential results is that proximal rewards are preferred by some groups more than others. For example, individuals from lower socio-economic backgrounds [74], and individuals with overweight and obesity may be more sensitive to rewards due to increased salience of the rewarding qualities of the stimulus [75]. As girls may be seen as a marginalised community within a school, especially with regard to physical activity [76], it is possible that this could be an explanation of why these components may have been particularly salient to girls. As suggested in our development work with adolescents, a sensitive approach to competition with private individual point totals like those used within GoActive may be worth further consideration.

Peer-leadership and online activity tracking were not associated with changes in MVPA or wellbeing, which is in contrast to previous evidence [77,78]. It should be noted that these intervention components may have potential for behaviour change, irrespective of our results. This is at least partly because there were implementation issues within and across participating schools that could have impacted effectiveness of different components. The difficulty of establishing what really went on during implementation is exacerbated by differences between process evaluation data obtained via focus groups, interviews, questionnaires and observations. Despite the questionnaire data reporting on the quality of peer-leadership, observations suggest that this component was rarely embraced by schools [25]. These implementation difficulties may speak to the need for beginning co-design and co-planning with schools, ideally on an individual institutional basis, before a project is even designed. Despite designing our intervention with students and teachers [20] and demonstrating effectiveness [79], the nature of a RCT requires testing this same intervention in schools who had not participated in the development process. As such, it is difficult to know how to appropriately test the effectiveness of a trial in the traditional clinical way, when, by definition, this means that schools cannot have as much input on the format of the intervention as they would like.

Although this is not a traditional mediation analysis, we tested our proposed logic model and examined the association between perception of intervention components and two important outcomes via a range of proposed psychosocial mediators [79]. The logic model was partly supported with mentorship, the most consistent component leading to change in wellbeing via a range of psychosocial mediators for both boys and girls. We also identified different patterns of mediation for boys and girls with teacher support and class sessions identified for boys and competition and rewards only for girls. This analysis provides insight for informing future interventions aiming to target wellbeing. Many interventions use components such as mentorship, leadership, class-based activity sessions and online activity tracking to increase physical activity; the potential for these physical activity interventions to additionally target wellbeing is becoming increasingly salient [62,80,81]. Despite this widespread use, relatively little is known about the mechanisms by which these intervention components may target outcomes via proposed mediators. This analysis provides valuable insight to identify promising intervention components and identifying mechanisms by which they may positively influence wellbeing.

### Summary in Relation to Participatory Co-Design Approach

Our results highlight several impasses between suggestions made by students in the co-design phase and then perception of components when implemented. In our process evaluation, students stated that they would have preferred the intervention being integrated into the school timetable [25], however, this directly contrasts with the suggestion in the co-design phase to have older students run the programme with distance from teachers and researchers [20]. Although participants indicated a desire to try non-standard activities in the development phase, when implemented, students were reluctant to choose and participate in unfamiliar activities, contrasting to the requests for novelty central to the participatory input in the earlier phase of the project [25]. Some components developed in the co-design phase were well liked, such as mentorship; mentorship was strongly suggested in the co-design phase and when mentorship was done well in the implementation phase, it was highly acceptable. The diverse range of opinions and preferences across individuals makes it challenging to incorporate multiple student ideas into programmes that can be implemented widely at scale. Although, in theory, intervention components such as mentorship and leadership align with student and teacher requests, in reality, the implementation of these components may not easily be incorporated in the school context. A different group of students from different schools participated in the co-design work to those that were participants in the randomised controlled trial; this could offer one explanation for why some components in the co-design process did not lead to change in physical activity and/or wellbeing. Conducting participant led-design with the same students that receive the intervention may overcome some of this incongruence but that appears problematic when aiming to implement a consistent programme at scale and evaluate it in a randomised controlled trial.

## 5. Strengths and Limitations

We recruited a population representative of the East of England and our results are relevant to many schools across the UK and to many other high-income settings. The percentage of pupils eligible for Pupil Premium in the participating schools was 20.9%, similar to the East of England average of 22.7% [81] and 86.1% of the recruited sample identified with White ethnicity, a similar proportion as England and Wales (87.4%) [82]. Our recruitment and retention to measurement sessions were high with 84% of eligible pupils measured at baseline; however, limitations include retention, although our percentage of valid data at follow-up is comparable to similar trials [19,83,84]. We studied intervention components comprising both perception and also online activity tracking data. It is a strength to focus on primary recipients’ experiences of and perspectives on interventions as participant perception of intervention components can help produce new insights regarding intervention design. A limitation is that mediators and outcomes were measured over the same time period, and so we do not know mediators’ preceded outcomes. There are potential unmeasured/unobserved confounders that were not included in this model (e.g., sedentary behaviour); however, we calculated the necessary size of unmeasured/unobserved to confound the effects through the sensitivity analysis of the e-value.

## 6. Conclusion

If implemented well, mentorship from older adolescents could have potential to increase social support, self-esteem and wellbeing among adolescents. If aiming to improve wellbeing, class-based sessions and perceived teacher support may be particularly beneficial among boys, whereas rewards and competition require further investigation for use to improve wellbeing amongst girls. No evidence was found to support the use of these components in school-based interventions to increase physical activity. Intervention components traditionally used to increase physical activity could have potential to be used in school-based interventions aiming to improve wellbeing.

## Figures and Tables

**Figure 1 ijerph-17-00390-f001:**
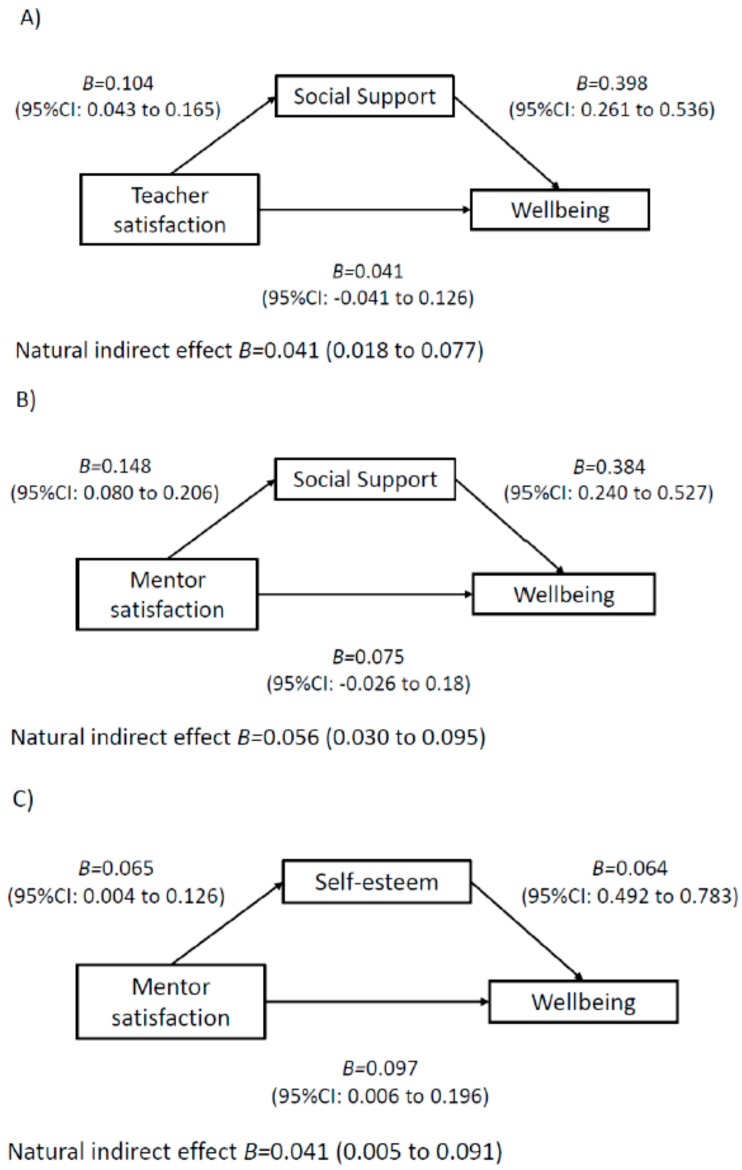
Mediation models of potential psychological mediators in the association between intervention components and wellbeing among **boys**. Note. Adjusted for age, ethnicity, language, school, BMI z-score and baseline values of change variables. CI, confidence interval. E-values estimates (in relative risk): Model **A**: E-value: 1.26, Model **B**: E-value: 1.33, Model **C**: E-value: 1.39.

**Figure 2 ijerph-17-00390-f002:**
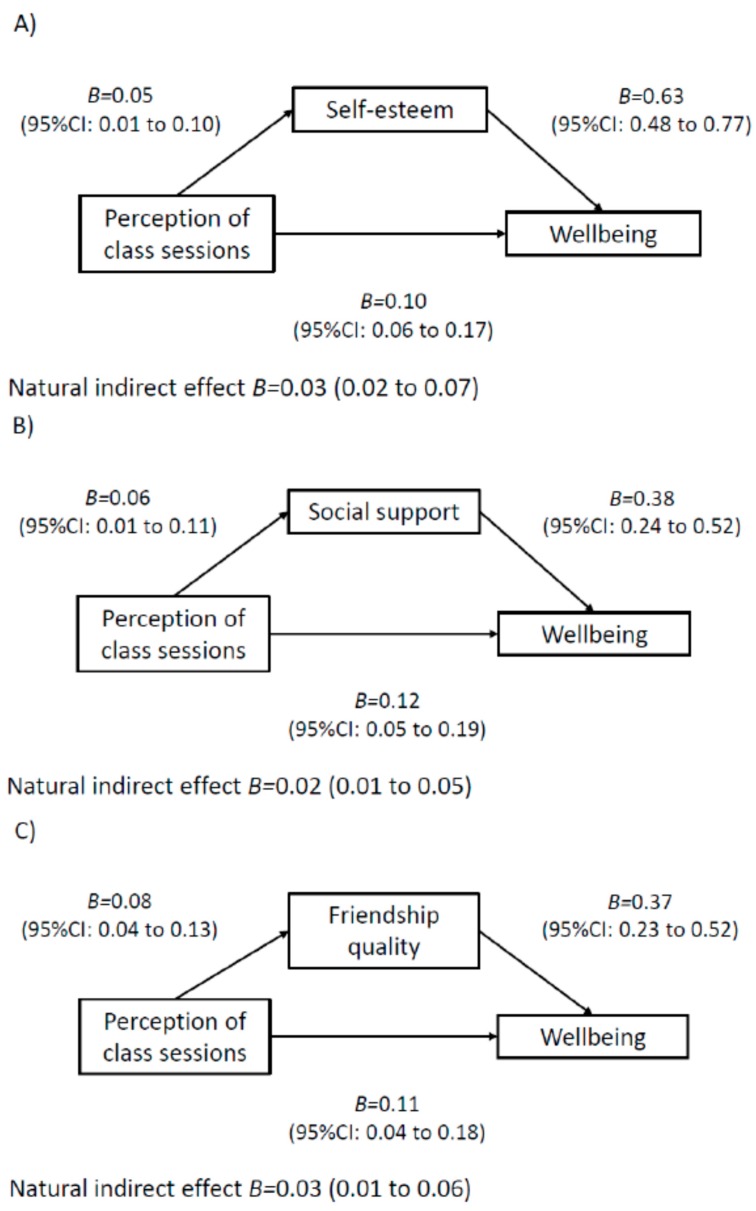
Mediation models of potential psychological mediators in the association between intervention components and wellbeing among boys. Note. Adjusted for age, ethnicity, language, school, body mass index and baseline values of change variables. CI, confidence interval. E-values estimates (in relative risk): Model **A**: E-value: 1.34, Model **B**: E-value: 1.29, Model **C**: E-value: 1.25.

**Figure 3 ijerph-17-00390-f003:**
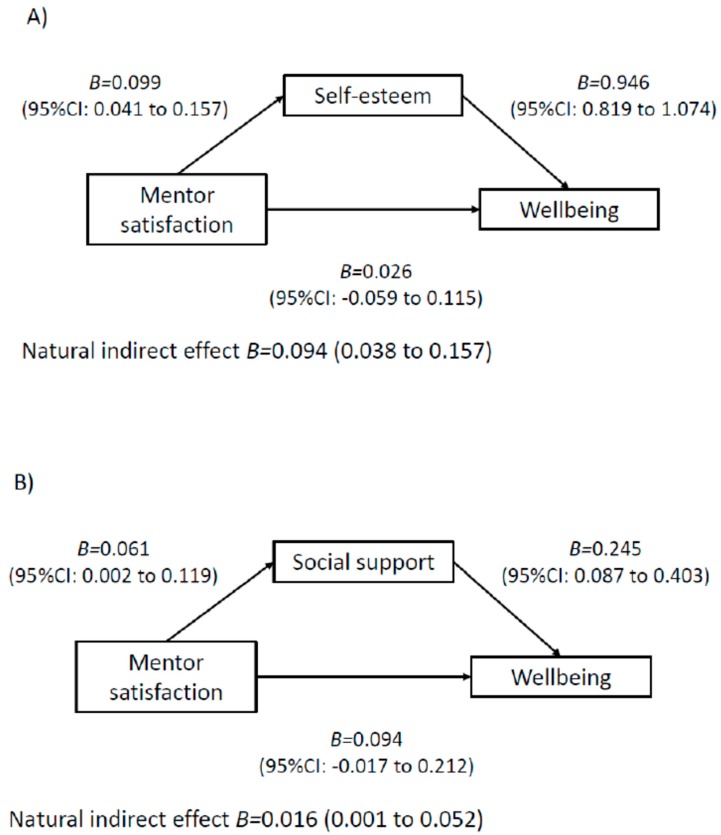
Mediation models of potential psychological mediators in the association between intervention components and wellbeing among girls. Note. Adjusted for age, ethnicity, language, school, BMI z-score and baseline values of change variables. CI, confidence interval. E-values estimates (in relative risk): Model **A**: E-value: 1.51, Model **B**: E-value: 1.16.

**Figure 4 ijerph-17-00390-f004:**
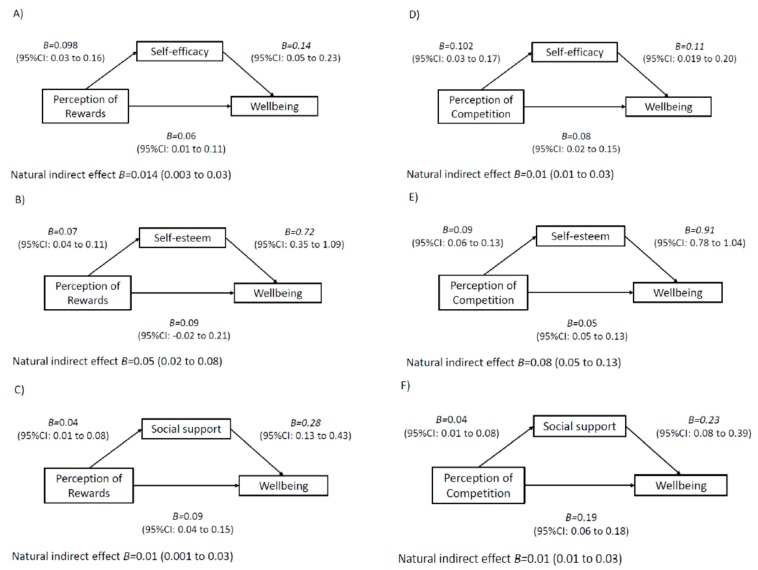
Mediation models of potential psychological mediators in the association between intervention components and wellbeing among girls. Note. Adjusted for age, ethnicity, language, school, BMI z-score and baseline values of change variables. CI, confidence interval. E-values estimates (in relative risk): Model **A**: E-value: 1.32, Model **B**: E-value: 1.19, Model **C**: E-value 1.33, Model **D**: 1.39, Model **E**: E-value 1.18, Model **F**: E-value 1.38.

**Table 1 ijerph-17-00390-t001:** Intervention components of the GoActive intervention with rationale from participatory development work.

Concept	Participatory Perspective Summary [20]	Component	Supporting Evidence
**Choice**	Adolescents identified that providing choice was important for Year 9 to be interested physical activity with the limited choice of school sports available considered to be a barrier to participation.	Each tutor group chooses two different activities weekly.	Adolescents given an activity choice have better programme attendance [27]. Choice may improve intrinsic motivation, self-efficacy and self-esteem, important for long-term activity maintenance [28,29].
**Novelty**	The small number of school sports available was a barrier to interest in physical activity and adolescents suggested introducing new types of activities. New activities were stated as important for reducing barriers regarding confidence and lack of skill in common sports as students would begin a new sport with equal ability.	There are 20 activities available, designed to utilise little or no equipment. Intervention materials are available on the study website, which include “quick-cards” (overviews of chosen activities).	Introducing adolescents to new activities is important; those given the opportunity to try new activities are more likely to want to do more [30].
**Mentorship**	Using older mentors or role models was suggested as more appealing than an intervention delivered by researchers or teachers. Participants suggested that these mentors should be slightly older but not too far from the participants’ age.	Older adolescents in the school (mentors) are paired with each Year 9 class and are responsible for encouraging their class to participate in new activities. Mentors are helped by Year 9 in-class leaders who change weekly.	Peers are crucial for adolescents to attain the best health behaviours in the transition to adulthood [31]. Cross-age mentorship can successfully improve adolescent health behaviours e.g., substance use [32,33], sexual health [34] and nutrition [35] but is understudied in physical activity research [36], particularly in young people [37].
**Competition**	Competition between tutor groups was suggested to promote participation among a whole school year group and to appeal to those students who would not normally get involved in physical activity. To encourage confidence, participants suggested private individual competition as well as class level competition. They suggested that the former should be kept private so as not to demotivate participants with lower scores. Teachers suggested that competition between tutor groups was an additional way to motivate teachers.	Students gain points every time they do an activity; there is no time limit, students just have to try an activity to get points. Individual points are kept private with class level totals announced to encourage inter-class competition. Students can enter their points on the GoActive website with individual passwords and login details.	Competitions improve engagement and retention in health promotion [38].
**Rewards**	Receiving rewards for certain levels of participation rather than performance were also suggested as motivating. This was thought to appeal to the competitive nature of students without emphasis on physical activity ability which may not appeal to less active participants.	Students gain small individual prizes for reaching certain points levels with everyone gaining a certain amount of points being entered into a prize draw for a bike.	Reward-based interventions appear effective in improving weight management behaviours in children [39].
**Flexibility**	There was no clear consensus about when was the best time for physical activity promotion with a range of times suggested, perhaps highlighting the need for flexibility within physical activity promotion. There was a lack of agreement regarding timing and location of activity, however, being able to participate with friends was considered important. Preferences for locations of activity also varied and highlighted the need for flexibility and choices that are sensitive to self-conscious adolescents.	During the feasibility and pilot work, one tutor time weekly has been used to do an activity and participants are also encouraged to do activities at other times, especially out of school.	A range of co-participants, timing and locations for activity are preferred by Year 9 adolescents with preferences differing on an individual level [30].
**Activity Sessions**	Teachers stated that time was an important barrier to teacher enthusiasm in physical activity interventions. Using tutor time (registration/roll call) physical activity promotion was suggested by teachers. Tutor time usually occurs first thing in the morning and after lunch at British schools when students attend a short class; their form tutor marks attendance and gives out school notices and reminders. Teachers could choose which tutor time(s) were used for running GoActive activities.	Each class was encouraged to use at least one tutor time weekly to participate in activities as a class together. In addition to during tutor time in the classroom, students were also encouraged to do activities at other times in and out of school. Some of the activities were group activities and some were individual. The full list of available activities is available as Appendix A.	Providing a new occasion to be active by replacing sedentary time for physical activity has been suggested to lead to successful physical activity promotion [40].

**Table 2 ijerph-17-00390-t002:** Descriptive characteristics participants included in analyses.

	Boys (*n* = 360)	Girls (*n* = 311)	P Value for Sex Difference
*Descriptive Characteristics*			
Baseline age (years)	13.23 (0.42)	13.24 (0.43)	0.966
Body mass index z-score	0.19 (1.25)	0.38 (1.15)	0.077
Language (only English), %	91.64	93.06	0.490
Ethnicity (white), %	84.89	87.50	0.327
*Outcomes*			
Moderate-to vigorous physical activity change (min/day)	−1.98 (23.40)	−1.55 (17.04)	0.901
Wellbeing change (score)	−0.03 (0.79)	−0.11 (0.72)	0.146
*Exposures*			
Perceived teacher support (score)	2.47 (0.93)	2.58 (0.91)	0.113
Perceived mentor support (score)	2.61 (0.83)	2.80 (0.77)	**0.002**
Web-based points entered (% versus not entered)	52.73	48.89	0.321
Perceived peer-leaders support (score)	0.47 (0.50)	0.36 (0.48)	**0.006**
Rewards	3.53 (1.17)	3.76 (1.35)	**0.024**
Competition	3.40 (1.07)	3.42 (1.25)	0.745
Class sessions	3.42 (1.16)	3.42 (1.24)	0.462
*Potential Mediators*			
Self-efficacy change (score)	−0.09 (0.91)	−0.10 (0.87)	0.955
Self-esteem change (score)	−0.02 (0.50)	−0.06 (0.45)	0.108
Social support change (score)	−0.11 (0.55)	−0.12 (0.46)	0.629
Group cohesion in-degree	−0.16 (1.37)	−0.28 (1.30)	0.159
Group cohesion out-degree	−0.05 (1.44)	−0.03 (1.24)	0.883
Friendship quality change (score)	−0.23 (0.55)	−0.21 (0.55)	0.990

Values are presented in percentage or mean and standard deviation. Bold text indicates that the confidence intervals cross zero.

**Table 3 ijerph-17-00390-t003:** Association between intervention components and potential mediators with outcomes.

	Boys		Girls	
	Physical Activity	Wellbeing	Physical Activity	Wellbeing
***Perception of Intervention Component***				
Teacher support	**2.93 (0.31 to 5.54)**	**0.08 (0.01 to 0.16)**	−0.50 (−2.41 to 1.43)	0.06 (−0.02 to 0.14)
Mentor support	1.47 (−1.51 to 4.45)	**0.14 (0.03 to 0.23)**	0.31 (−1.87 to 2.50)	**0.11 (0.02 to 0.20)**
Class sessions	1.83 (−0.31, 3.96)	**0.10 (0.03, 0.18)**	0.20 (−1.18, 1.57)	0.04 (−0.02, 0.05)
Peer-leadership	−4.42 (−9.25 to 0.41)	0.11 (−0.04 to 0.25)	−0.91 (−4.56 to 2.74)	−0.09 (−0.23 to 0.06)
Rewards	**2.53 (0.35 to 4.71)**	**0.07 (0.01 to 0.14)**	0.35 (−0.97 to 1.67)	**0.10 (0.05 to 0.15)**
Competition	1.26 (−1.16 to 3.67)	0.06 (−0.02 to 0.14)	0.53 (−0.87 to 1.92)	**0.13 (0.07 to 0.18)**
***Online Intervention Component***				
Web-based points	−0.04 (−4.79 to 4.71)	0.06 (−0.09 to 0.20)	−1.74 (−5.15 to 1.67)	0.06 (−0.07 to 0.19)
***Potential Mediators***				
Self-efficacy	−1.10 (−3.94 to 1.75)	0.08 (−0.01 to 0.16)	1.75 (−0.32 to 3.82)	**0.20 (0.12 to 0.28)**
Self-esteem	4.19 (−1.25 to 9.63)	**0.66 (0.51 to 0.80**	1.22 (−2.71 to 5.15)	**0.95 (0.83 to 1.08)**
Social support	−2.90 (−7.51 to 1.71)	**0.43 (0.29 to 0.56)**	1.25 (−2.47 to 4.98)	**0.28 (0.14 to 0.42)**
Friendship quality	4.86 (−0.05 to 9.76)	**0.42 (0.28 to 0.57)**	2.29 (−1.19 to 5.78)	**0.66 (0.53 to 0.78)**
Group cohesion in-degree	0.65 (−1.42 to 2.71)	−0.01 (−0.08 to 0.07)	0.08 (−1.44 to 1.60)	−0.01 (−0.07 to 0.06)
Group cohesion out-degree	−0.87 (−3.00 to 1.26)	0.04 (−0.03 to 0.12)	−0.38 (−2.08 to 1.33)	0.04 (−0.03 to 0.11)

Note. Values are presented using unstandardized coefficients and 95% confidence intervals. Adjusted for age, ethnicity, language, school, body mass index. Bold text indicates that the confidence intervals cross zero.

**Table 4 ijerph-17-00390-t004:** Association between perception of intervention components and potential mediators.

	Self-Efficacy	Self-Esteem	Social Support	Friendship Quality	GC In-Degree	GC Out-Degree
**Boys**						
Teacher support	**0.12 (0.01 to 0.22)**	0.02 (−0.04 to 0.07)	**0.09 (0.03 to 0.15)**	0.00 (−0.05 to 0.06)	−0.14 (−0.29 to 0.01)	0.01 (−0.14 to 0.15)
Mentor support	**0.15 (0.04 to 0.27)**	**0.07 (0.01 to 0.13)**	**0.14 (0.07 to 0.21)**	0.06 (−0.01 to 0.13)	−0.09 (−0.26 to 0.09)	0.12 (−0.05 to 0.28)
Class sessions	0.02 (−0.07 to 0.11)	**0.05 (0.01 to 0.10)**	**0.05 (0.01 to 0.10)**	**0.09 (0.04 to 0.13)**	0.01 (−0.12 to 0.13)	**0.20 (0.08 to 0.31)**
Peer-leadership	0.15 (−0.05 to 0.34)	−0.08 (−0.18 to 0.02)	0.06 (−0.05 to 0.17)	−0.04 (−0.15 to 0.08)	−0.22 (−0.50 to 0.53)	0.14 (−0.14 to 0.41)
Rewards	0.06 (−0.03 to 0.14)	0.01 (−0.04 to 0.06)	0.04 (−0.01 to 0.09)	0.04 (−0.02 to 0.09)	−0.07 (−0.21, 0.08)	0.01 (−0.15, 0.16)
Competition	0.02 (−0.07 to 0.12)	−0.02 (−0.07 to 0.03)	0.04 (−0.02 to 0.10)	0.04 (−0.01 to 0.08)	−0.14 (−0.29, 0.02)	−0.02 (−0.19, 0.14)
Web-based points	0.11 (−0.09 to 0.31)	−0.03 (−0.13 to 0.07)	0.06 (−0.05 to 0.17)	−0.04 (−0.15 to 0.07)	0.25 (−0.02 to 0.52)	**0.32 (0.06 to 0.58)**
**Girls**						
Teacher support	0.06 (−0.03 to 0.16)	0.04 (−0.01 to 0.09)	0.03 (−0.01 to 0.08)	0.05 (−0.01 to 0.11)	**−0.16 (−0.02 to −0.29)**	−0.05 (−0.17 to 0.06)
Mentor support	0.10 (−0.01 to 0.20)	**0.11 (0.05 to 0.17)**	**0.06 (0.01 to 0.12)**	−0.01 (−0.08 to 0.06)	**−0.21 (−0.04 to −0.37)**	−0.01 (−0.16 to 0.14)
Class sessions	0.05 (−0.02, 0.12)	**0.07 (0.03 to 0.10)**	**0.06 (0.02 to 0.10)**	**0.05 (0.01, 0.09)**	0.03 (−0.07, 0.13)	0.04 (−0.05 to 0.13)
Peer-leadership	0.04 (−0.14 to 0.22)	−0.01 (−0.10 to 0.09)	0.00 (−0.10 to 0.10)	−0.02 (−0.13 to 0.09)	−0.16 (−0.43 to 0.11)	−0.07 (−0.31 to 0.17)
Rewards	**0.10 (0.03 to 0.16)**	**0.08 (0.04 to 0.11)**	**0.04 (0.01 to 0.07)**	**0.04 (0.01 to 0.08)**	−0.07 (−0.17, 0.04)	0.06 (−0.04, 0.16)
Competition	**0.10 (0.03 to 0.17)**	**0.10 (0.06 to 0.13)**	**0.04 (0.01 to 0.08)**	0.03 (−0.02 to 0.09)	−0.04 (−0.15, 0.08)	**0.12 (0.01, 0.23)**
Web-based points	0.12 (−0.04 to 0.28)	0.00 (−0.09 to 0.09)	0.01 (−0.11 to 0.08)	0.02 (−0.08 to 0.12)	−0.06 (−0.30 to 0.19)	0.01 (−0.21 to 0.23)

Values are presented using unstandardized coefficients and 95% confidence intervals. Adjusted for age, ethnicity, language, school, BMI z-score and baseline values. GC Group cohesion. Bold text indicates that the confidence intervals cross zero.

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
