# Peer review of "Pathways to Increasing Adolescent Physical Activity and Wellbeing: A Mediation Analysis of Intervention Components Designed Using a Participatory Approach"

_ijerph, 2020, doi:10.3390/ijerph17020390_

Round 1

Reviewer 1 Report

Please present in Introduction more recent studies in this direction and most important for this chapter which is the recommendations of WHO about practicing PA for adolescents (see recommendations levels of PA for children aged 5-17 years). Also, please present which is the novelty of this study.

I want to ask the authors that the students have hours in July? Which is the structure of school year? Please detailed this......

The authors must detailed the questionnaire WEMWS about who validated this, the value of alpha cronbach, and what was the interpretation value of this.

Please rearrange the reference in accordance with the policy of journal.

Author Response

We thank the Academic Editor and reviewers for their thoughtful comments and have made a point by point response to the suggestions below and they have been highlighted in red in the revised version of the manuscript. We believe that the manuscript is much improved due to edits made in response to these comments.

Reviewer 1

Please present in Introduction more recent studies in this direction and most important for this chapter which is the recommendations of WHO about practicing PA for adolescents (see recommendations levels of PA for children aged 5-17 years).

Response:

We have added to the introduction regarding the WHO recommendations and also the recent physical activity guidelines. We have added some more references to studies examining physical activity levels in adolescents as follows:

Globally, physical inactivity is thought to cause 9% of premature deaths and is estimated to cost 53.8 billion in health care [2,3]. Due to the importance of inactivity as a health risk and the high prevalence of inactivity worldwide, one of the World Health Organisation’s nine global targets is a 10% relative reduction in the prevalence of inactivity by 2025 [4]. However, recent global data suggests that meeting this global target looks increasingly unlikely [5]. The World Health Organisation recommend that all children between 5 and 17 years-old should do at least 60 minutes of physical activity every day [6] which aligns with British recommendations [7]. Recent evidence suggests that worldwide, the majority of young people (81%) aged 11 to 17 years do not meet these recommendations [5].

We have also extended the introduction and added references regarding wellbeing.

Risk factors for poor mental health consist or a broad range of individual, family, environmental, social and other factors [12]; this range of correlates is likely to contribute to the large variation in efficaciousness in trials aiming to improve mental health in adolescence [15]. More universally effective strategies are needed to support and enable positive mental health, including wellbeing [16].

Also, please present which is the novelty of this study.

Response:

We have clarified the novelty of the studies in the last paragraph of the introduction as follows:

“We aim to apply mediation analysis in a novel approach to evaluate the potential mediating role of psycho-social factors (social support, self-efficacy, group cohesion, friendship quality and self-esteem) in the association between engagement in intervention components suggested by students in our intervention co-design process (mentorship, leadership, class-based activity sessions, competition and rewards) with changes in physical activity and wellbeing.”

I want to ask the authors that the students have hours in July? Which is the structure of school year? Please detailed this......

Response:

Schools in the UK usually break up for summer holidays in the third week of July. The following has been added to the methods section to explain the structure of the school year in England.

“The school year in British non-fee paying schools usually runs from early September to the third week of July. There are holidays at Christmas and Easter (approximately two weeks each) with ‘half-term’ (one week holidays) in late October, mid/late February and late May.”

The authors must detailed the questionnaire WEMWS about who validated this, the value of alpha cronbach, and what was the interpretation value of this.

Response:

More information about the WEMWS has been added, including Cronbach’s alpha, as follows:

“Items relate to both hedonic and eudaimonic experiences of mental health including positive affect (e.g., ‘I’ve been feeling optimistic about the future’), relationships (e.g., ‘I’ve been feeling close to other people’) and emotional functioning (e.g., ‘I’ve been dealing with problems well’),…”

“The scale has shown good content validity, and correlates well with other mental health and wellbeing scales including the Positive and Negative Affect Scale [30,31], Short Depression-Happiness Scale [32] and the World Health Organisation-Five Well-Being Index [33]. Cronbach’s alpha has been shown to be 0.89 among a student sample and 0.91 in a population sample [29].”

Please rearrange the reference in accordance with the policy of journal.

Response:

We have downloaded the Endnote template from the journal website and used that to format our references. We found one mistake which we have corrected.

Reviewer 2 Report

The description of the research method does not indicate the types of proposed forms of physical activity. Were they individual or team forms? Where did those classes take place? Were they held during the week after school or during the weekend? Were they organized for entire classes?

The conclusions did not show any connections between physical activity and other components of the research.

Author Response

We thank the Academic Editor and reviewers for their thoughtful comments and have made a point by point response to the suggestions below and they have been highlighted in red in the revised version of the manuscript. We believe that the manuscript is much improved due to edits made in response to these comments.

Reviewer 2

The description of the research method does not indicate the types of proposed forms of physical activity.

Response:

There were 19 types of activity available for students to choose, this was in response to co-design work with students who suggested the need to appeal to many individuals with different preferences. A table (new Table 1) has been added to describe the intervention in further detail and to address the content-related comments of this reviewer. This table also includes a summary of why this element was included, with regards to the participatory development process as requested by the academic editor. Responses to all of the reviewer’s questions are now included in Table 1 and we have summarized the responses below.

Were they individual or team forms?

Some of the activities were group activities and some were individual. The full list of the 19 available activities has been uploaded as supplementary material.

Where did those classes take place?

In addition to during tutor time in the classroom at least once per week, students were also encouraged to do activities at other times in and out of school.

Were they held during the week after school or during the weekend?

Each class was encouraged to use at least one tutor time weekly to participate in activities as a class together. Tutor time usually occurs first thing in the morning and after lunch at British schools when students attend a short class; their form tutor marks attendance and gives out school notices and reminders. Teachers could choose which tutor time(s) were used for running GoActive activities. In addition to during tutor time in the classroom, students were also encouraged to do activities at other times in and out of school.

Were they organized for entire classes?

Each class was encouraged to use at least one tutor time weekly to participate in activities as a class together.

The conclusions did not show any connections between physical activity and other components of the research.

Response:

The authors have added to the conclusion that no associations were seen for physical activity in this paper as follows:

“No evidence was found to support the use of these components in school-based interventions to increase physical activity.”

The following has also been added to the abstract:

“No evidence was found to support the use of these components to increase physical activity”

Round 2

Reviewer 1 Report

Please arrange the article in accordance with the policy of journal, see template on the site of journal.